# Efficient Combination of Complex Chromatography, Molecular Docking and Enzyme Kinetics for Exploration of Acetylcholinesterase Inhibitors from *Poria cocos*

**DOI:** 10.3390/molecules28031228

**Published:** 2023-01-27

**Authors:** Tong Wu, Wanchao Hou, Chunming Liu, Sainan Li, Yuchi Zhang

**Affiliations:** 1College of Pharmacy, Changchun University of Chinese Medicine, No. 1035 Jingyue Street, Nanguan District, Changchun 130117, China; 2College of Pharmacy, Jilin University, No. 2699 Qianjin Road, Chaoyang District, Changchun 130012, China; 3Central Laboratory, Changchun Normal University, No. 677 North Changji Road, Erdao District, Changchun 130032, China

**Keywords:** *Poria cocos*, acetylcholinesterase inhibitors, molecular docking technique, enzymatic reactions kinetics, counter-current chromatography, semi-preparative high−performance liquid chromatography

## Abstract

*Poria cocos (P. cocos)* is a traditional Chinese medicinal product with the same origin as medicine and food. It has diuretic, anti-inflammatory and liver protection properties, and has been widely used in a Chinese medicine in the treatment of Alzheimer’s disease (AD). This study was conducted to explore the activity screening, isolation of acetylcholinesterase inhibitors (AChEIs), and in vitro inhibiting effect of *P. cocos*. The aim was to develop a new extraction process optimization method based on the Matlab genetic algorithm combined with a traditional orthogonal experiment. Moreover, bio−affinity ultrafiltration combined with molecular docking was used to screen and evaluate the activity of the AChEIs, which were subsequently isolated and purified using high-speed counter−current chromatography (HSCCC) and semi−preparative high-performance liquid chromatography (semi−preparative HPLC). The change in acetylcholinesterase (AChE) activity was tested using an enzymatic reaction kinetics experiment to reflect the inhibitory effect of active compounds on AChE and explore its mechanism of action. Five potential AChEIs were screened via bio−affinity ultrafiltration. Molecular docking results showed that they had good binding affinity for the active site of AChE. Meanwhile, the five active compounds had reversible inhibitory effects on AChE: Polyporenic acid C and Tumulosic acid were non-competitive inhibitors; 3−Epidehydrotumulosic acid was a mixed inhibitor; and Pachymic acid and Dehydrotrametenolic acid were competitive inhibitors. This study provided a basis for the comprehensive utilization of *P. cocos* and drug development for the treatment of AD.

## 1. Introduction

*Poria cocos* (*P. cocos*) is a well-known traditional East Asian medicinal plant that grows around the roots of pine trees in China, Japan, Korea, and North America [1,2]. Through research, summary, and analysis of the current status of traditional Chinese medicine in the treatment of Alzheimer’s disease (AD), Gong Jing found that the use of *P. cocos* ranks among the top 10 most commonly used unilateral drugs [3]. Hu Pan discussed the ancient medication rules for AD based on the traditional Chinese medicine inheritance computing platform. Of these, the core drug combination "*Poria cocos*-Licorice" has a good effect on the treatment of AD [4]. Zhou Siduo used an improved micro-screening model based on the Ellman method to screen 49 kinds of traditional Chinese medicine extracts for acetylcholinesterase (AChE) activity and calculate the inhibition rate. The results showed that *P. cocos* had a better ability to inhibit AChE [5].

Alzheimer’s disease (AD) is the most common cause of dementia in elderly people. Loss of brain cholinergic function is an established neuropathological observation in this disease [6]. Therefore, acetylcholinesterase inhibitors (AChEIs) are widely used for the symptomatic treatment of AD patients. Their mode of action primarily comprises the inhibition of AChE activity, increasing and accumulating acetylcholine in the brain, and enhancing neuronal transmission to prevent and treat AD [7]. Medicinal plants are relatively safe and inexpensive. Therefore, they represent a good alternative for potential new anti-AD compounds. Natural products and phytomedicines are the biggest hope for curing various human degenerative diseases, including AD. Plants have also been investigated as a source of medicinally relevant bioactive components; *P. cocos* appear to have a strong therapeutic value in the treatment of AD.

High−speed counter−current chromatography (HSCCC) is a chromatographic separation technology developed in the 1980s which is based on liquid-liquid extraction. HSCCC is widely used in the separation and preparation of natural products due to its large preparation volume, fast separation speed, irreversible adsorption, and high sample recovery compared with traditional chromatography [8,9]. Semi−preparative high performance liquid chromatography (Semi−preparative HPLC), a classical liquid-solid separation method that exhibits excellent reproducibility and facile operation, is used extensively in the separation and purification of natural products [10,11]. Because of the similar polarity of the chemical components in *P. cocos*, the separation result of the single separation method is not ideal. Therefore, this study used the complementary advantages of the two methods to separate the active components in *P. cocos*. 

Three types of activity methods were used to study potential AChEIs in *P. cocos*. Compounds from plants with inhibitory activities were conventionally screened using in vitro assays [12]. A method based on ultrafiltration–liquid chromatography combined with mass spectrometry (UF−LC−MS) has been proposed to overcome the limitations of in vitro methods and enhance the throughput of drug discovery [13,14]. UF−LC−MS is a powerful tool for screening and isolating biologically active compounds from botanical extracts. It is a combination of affinity ultrafiltration and mass spectrometry. In the ultrafiltration step, the ligand−receptor complex can be separated from the unbound small molecules, and the active small molecules in the complex can be released by adding an organic solvent. The ligands can then be identified by the LC−MS step, and the active small molecules can be initially identified [15]. Molecular docking was used to simulate the binding of the active components of the monomer to AChE. Based on the results obtained after docking, potential proteins and specific compounds with targeted effects were identified to determine the interaction between active ingredients with disease-related target proteins and their stable binding ability [16,17,18]. The molecular docking virtual screening method improves the efficiency of chemical activity evaluation and the discovery direction of lead compounds, providing a new technique for mining traditional Chinese medicinal resources [19]. Meanwhile, a high−performance capillary electrophoresis (HPCE) method based on AChE was established by combining HPCE with enzymatic reaction kinetics, which is used in the investigation of the inhibition types of active ingredients, as well as to explore their inhibition mechanism [20,21,22,23,24]. This method exhibits several advantages, such as high efficiency, short analysis time, various separation methods, a high degree of automation, and a small sampling volume [25,26]. 

In this study, the active ingredients from *P. cocos* were extracted using the heat reflux method. Taking AChE as the biological target molecule, the active ingredients were screened by UF−LC−MS, and the results were verified from a molecular point of view using molecular docking technology. Active components were separated using HSCCC combined with semi-preparative HPLC. The inhibition mechanism of the active ingredients was studied using HPCE combined with enzymatic reaction kinetics. A method for identifying the composition structure based on chromatography−mass spectrometry analysis and a separation method of active ingredients with high efficiency and high recovery was established. The experimental process is shown in Figure 1.

## 2. Results

### 2.1. Optimization of Extraction of P. cocos Reflux

The ethanol reflux extraction process of *P. cocos* was optimized using the Matlab genetic algorithm combined with an orthogonal experiment. The quadratic regression model of total triterpene content was used as the objective function, and the optimal extraction conditions were searched by the genetic algorithm. The initial population size was 50, the single point mutation probability was 0.01, the single point crossover probability was 0.85, the evolution was 100 generations, and the results were searched randomly 10 times. The optimal extraction conditions were 19.87 times the ethanol dosage, 1.06 h of extraction time, 1.09 h extraction time, and 80% ethanol concentration. Under these conditions, the theoretical value of Pachymic acid content reached 0.83%. For operability of the experimental conditions, the extraction conditions were further modified to 20 times the ethanol dosage, 1.0 h extraction time, and 80% ethanol concentration. Under the optimized conditions, the active components were extracted from *P. cocos*. The eluent was condensed at 45 °C by rotary vaporization to a dry state, and the concentrate was evaporated and transformed into a powder for subsequent use.

### 2.2. Screening and Evaluation of Potential AChE Inhibitors in P. cocos

#### 2.2.1. Screening of Potential AChEIs in *P. cocos* by UF-LC-MS

According to the chromatographic results of the ligand (potential AChEIs) in *P. cocos*, in the range of 25–55 min, 12 components were eluted from *P. cocos* extract, five of which exhibited an affinity to the AChE receptor. The results were confirmed using AChE at varying levels of activity. As illustrated in Figure 2, the retention time of compound 1 was 36.144 min, and the inhibition rates of 0.5, 1.0, and 2.0 U/mL AChE were 24.10, 24.34, and 24.22%, respectively; For compound 2, the retention time was 39.562 min, and the inhibition rates at 0.5, 1.0, and 2.0 U/mL AChE were 36.59, 37.46, and 37.32%, respectively; For compound 3, the retention time was 41.032 min, and the inhibition rates at 0.5, 1.0, and 2.0 U/mL AChE were 13.12, 13.88, and 13.78%, respectively; For compound 4, the retention time was 46.066 min, and the inhibition rates at 0.5, 1.0, and 2.0 U/mL AChE were 38.32, 38.60, and 38.52%, respectively; For compound 5, the retention time was 55.058 min, and the inhibition rates at 0.5, 1.0, and 2.0 U/mL and AChE were 11.87, 12.03, and 11.90%, respectively. The mixture of *P. cocos* extract with 1.0 U/mL AChE exhibited the highest binding degree. The binding strength of the compound and AChE was determined as expressed in Equation (1). The order of combined strength capabilities of the active ingredient and AChE was 4 (38.60%) > 2 (37.46%) > 1 (24.34%) > 3 (13.88%) > 5 (12.03%). The five active ingredients that were identified by mass spectrometry were (1) Tumulosic acid, (2) Polyporenic acid C, (3) 3-Epidehydrotumulosic acid, (4), Pachymic acid, and (5) Dehydrotrametenolic acid.

#### 2.2.2. Molecular Docking Simulation of Active Compounds and AChE

The active pocket of AChE mainly contains two ligand active binding sites; one of which is the acylation site and the other is the peripheral anion binding site. The acylation site is near the bottom of the pocket and contains a very important residue, SER (serine). The peripheral anion binding site is located at the edge of the pocket and contains two important residues, the aromatic residue TRP (tryptophan) and the negatively charged residue ASP (Aspartic). The active sites of AChE were analyzed by ligand expansion and SiteMap search. The ligand expansion method was used to define the amino acid within 0.5 nm of the eutectic compound in the crystal structure of acetylcholinesterase as the active site.

The complex conformation of AChE with Tumulosic acid, Polyporenic acid C, 3−Epidehydrotumulosic acid, Pachymic acid, and Dehydrotrametenolic acid was simulated at 298 K using an Amber99sb−ILDN force field. The crystal structure of human AChE (PDB code: 2ACK) was downloaded from the Protein Database. Before the molecular docking experiment, Autodock Tool 1.5.4 software was used to hydrogenate the protein and calculate the Gasteiger charge. Non−polar hydrogen atoms were combined to determine the atomic type, and the non-integer charge on the amino acid residues in AChE was corrected. Key potential target proteins were downloaded from the PDB database. PyMOL software was used for the pretreatment of protein receptor molecules, and Chemdraw 3D software was used for the pretreatment of active compound ligands. The pre-processed potential targets and active compounds were imported into the docking software AutoDock Vina 1.2.0 for molecular docking, and the results were saved in pdbqt format. Huperzine A is the AChEIs extracted from natural products with the best clinical effect in the treatment of AD. Therefore, it was simulated with the target protein, and the binding ability of five active components with AChE in *P.cocos* was compared by the binding energy of the dock binding energy. Coordinates of the central grid point of maps parameter setting: Pachymic acid: (x, y, z) = (4.4, 68.5, 63.0); Polyporenic acid C: (x, y, z) = (4.36, 68.4, 62.5); Tumulosic acid: (x, y, z) = (4.43, 67.1, 63.4); 3-Epidehydrotumulosic acid: (x, y, z) = (4.39, 68.37, 63.3); Dehydrotrametenolic acid: (x, y, z) = (4.4, 68.51, 64.3); and Huperzine A: (x, y, z) = (4.5, 66.71, 63.3). In this process, a dodecahedron was used as the simulation system, TIP3P was used as the filled water molecule model, and Na^+^ was used as the counterbalance ion to balance the system charge. The results are shown in Figure 3. 

The results showed that Huperzine A and Pachymic acid only acted on the acylating active site of AChE, Tumulosic acid and 3−Epidehydrotumulosic acid only acted on the peripheral anion acting site of AChE, and Polyporenic acid C and Dehydrotrametenolic acid could simultaneously act on both the active site and the peripheral anion acting site of AChE. The average binding energies of the active Tumulosic acid (1), Polyporenic acid C (2), 3−Epidehydrotumulosic acid (3), Pachymic acid (4), Dehydrotrametenolic acid (5), and Huperzine A (6) were −7.77 kcal/mol, −7.88 kcal/mol, −7.37 kcal/mol, −8.01 kcal/mol, −6.92 kcal/mol and −7.51 kcal/mol, as shown in Appendix A. The larger the absolute value of the binding energy, the better the binding effect and the stronger the inhibition. These results demonstrated that the model effect is good, and it is consistent with the results of the ultrafiltration experiments, which further verifies the reliability of the ultrafiltration experiments.

### 2.3. Isolation of Active Compounds from P. cocos

#### 2.3.1. HSCCC Separation of Potential AChE Inhibitors from *P. cocos*

According to the conditions described in Section 4.7.1, the active components were separated from *P. cocos* ethanol extract, as shown in Figure 4. When we used PET : EtOAc : MeOH : H_2_O (4.0:1.0:3.0:2.0, *v/v/v/v*) as the solvent system, the overall elution time was approximately 350 min; after 270 min, the baseline tended to level off. Three clear chromatographic peaks observed in the spectrogram corresponded to the active compounds 2, 3, and 4. HPLC was used to determine the purity of the isolated active ingredients, and the purities of the four components were 2 (96.29%), 3 (95.40%) and 4 (97.31%). The purity of the three active ingredients is higher than 95%, which should be considered during follow−up research.

#### 2.3.2. Semi−Preparative HPLC Separation of Potential AChEIs from *P. cocos*

Both HSCCC and semi−preparative HPLC can be used to isolate an active compound from purified substances; however, the separation abilities of the two methods vary. Therefore, on the basis of separation by HSCCC, semi-preparative HPLC was used to further separate the active components of *P. cocos*. The separation conditions are discussed in Section 4.7.2. Two high−purity monomer compounds were eluted using semi−preparative HPLC, as shown in Figure 5. HPLC detected active compounds 1 and 5, and the purities of the active compounds were 98.11% and 97.82%, respectively.

### 2.4. Structural Identification of Active Compounds

The potential AChE inhibitors in *P. cocos* extract were analyzed using UPLC−Q−Exactive−MS. The five bioactive components in *P. cocos* were isolated using HSCCC and semi−preparative HPLC. The chromatograms are shown in Figure 6, and the purity of the four target compounds exceeded 95%. The HPLC determined purities of compounds 1, 2, 3, 4, and 5 were 98.11%, 96.29%, 95.40%, 97.31%, and 97.82%, respectively. The high-purity active ingredients were than processed for the next enzymatic reaction kinetic study.

#### 2.4.1. Identification of Potential AChEIs Using UPLC−Q−Exactive−MS

The UPLC−Q−Exactive method was used to identify five isolated parts. All five active components exhibited strong signals in negative ion mode. Peaks were assigned based on the HPLC retention times and MS/MS data (Table 1). MS/MS data were obtained and compared with previously reported data to identify the five compounds. The experimentally obtained [M−H]^−^ (*m/z*) values of compounds 1, 2, 3, 4 and 5 were 485.2734, 481.3371, 483.7835, 527.3791, and 453.3451, respectively. By comparing the MS data and LC retention times with those of the standards, target compounds 1, 2, 3, 4 and 5 were identified as Tumulosic acid, Polyporenic acid C, 3−Epidehydrotumulosic acid, Pachymic acid, and Dehydrotrametenolic acid, respectively [18,27,28,29,30].

#### 2.4.2. Identification of Potential AChEIs Using ^1^H-NMR and ^13^C−NMR Spectroscopy

The ^1^H−NMR and ^13^C−NMR data were in good agreement with those of the corresponding compounds. The main features of compounds 1, 2, 3, 4 and 5 are summarized in Appendix A.

### 2.5. Study on Enzymatic Reaction Kinetics of the Inhibition of AChE by Active Compounds

According to the electroswimming strip introduced in 4.7.2, an electroswimming diagram of the standard mixture solution is obtained (Figure 7). It can be observed that acetylcholine (ACh) and choline (Ch) achieved adequate separation within 20 min, and the separation reoccurrence was excellent.

#### 2.5.1. Inhibitory Types of Active Compounds in *P. cocos* on AChE

The effects of 10 mmol/L changes in the concentration of AChE (500, 1000, 1500, 2000, and 2500 U/L) determined the enzyme activity by changing the concentration of AChE (100, 300, 500, 800, and 1000 μmol/L). The results are illustrated in Figure 8(1(a), 2(a), 3(a), 4(a), and 5(a)). Under the conditions of monomer compounds with varying concentrations, all lines pass through the origin, and with the increase in inhibitor concentration, the slope of the lines gradually decreases, indicating that the inhibition types of monomer compounds on AChE are all reversible.

#### 2.5.2. Inhibition Kinetics Analysis of Active Compounds in *P*. *cocos* on AChE

The concentration of AChE was fixed at 1000 U/L, and the concentration of ACh was changed to determine the effects of varying concentrations (0, 10.0, 15.0, and 20.0 μmol/L) on enzyme activity. The reaction rates at substrate concentrations of 5, 10, 15, and 20 mmol/L were measured and plotted using the double reciprocal, with 1/v as the ordinate and the reciprocal of 1/[S] as the abscissa, and the results are shown in Figure 8 (1(b), 2(b), 3(b), 4(b), 5(b)). The results showed that the intersection of the double-reciprocal curves of inhibitors Pachymic acid and Dehydrotrametenolic acid with different concentrations was located on the Y−axis, which suggested that they were competitive inhibitors of AChE. The intersection of the double−-reciprocal curves of Polyporenic acid C and Tumulosic acid inhibitors with different concentrations was located on the X−axis, suggesting that they were non-competitive inhibitors of AChE. The intersection point of the double reciprocal curve of inhibitor 3−Epidehydrotumulosic acid with different concentrations was found in the quadrant, so it was speculated that it was a mixed inhibitor of AChE. Since the intersection point is located in the second quadrant, it is further speculated to be non-competitive–competitive inhibition. The Michaelis–Menten equation and the kinetic parameters of AChE inhibition of active ingredients are shown in Appendix A.

## 3. Discussion

Current clinical drugs used to treat AD inhibit AChE activity in the body, preventing the degradation of ACh in the synapses and increasing the number of nerve cells in ACh. Therefore, the screening of AChEIs is key to the drug development process. In addition, traditional Chinese medicine is often the source of new active compounds; therefore, screening AChEIs from traditional Chinese medicine is an important trend in AD drug development.

Five potential AChEIs were screened from the alcoholic extract of *P. cocos* by UF−LC−MS, and their binding strengths were Tumulosic acid (24.34%), Polyporenic acid C (37.46%), 3−Epidehydrotumulosic acid (13.88%), Pachymic acid (38.60%), and Dehydrotrametenolic acid (2.03%). Using molecular docking technology, the active compound and the target protein AChE were docked and simulated, and the average binding energies were −7.77 kcal/mol, −7.88 kcal/mol, −7.37 kcal/mol, −8.01 kcal/mol, and −6.92 kcal/mol, respectively. The larger the absolute value of the binding energy, the better the binding effect and the stronger the AChE inhibitory ability. Therefore, the results of molecular docking show that the inhibitory ability of the five active components of AChE is as follows: Pachymic acid > Polyporenic acid C > Tumulosic acid > 3−Epidehydrotumulosic acid > Dehydrotrametenolic acid, which is consistent with the ultrafiltration results, which further verifies the accuracy of the ultrafiltration experimental results. UF−LC−MS coupled with molecular docking is a powerful method for screening biologically active compounds in botanical extracts.

The five active ingredients were successfully separated by HSCCC combined with semi−preparative HPLC, and the purities were 98.11%, 96.24%, 97.31%, 99.17%, and 97.82%, respectively. The results show that this method has a good effect on the separation of chemical components in *P. cocos*. The inhibition types and mechanisms of the active ingredients in AChE were investigated by combining HPCE with enzymatic reaction kinetics. The results showed that the five active ingredients were all reversible inhibitors, among which Polyporenic acid C and Tumulosic acid were non-competitive inhibitors of AChE, 3−Epidehydrotumulosic acid was a mixed inhibitor of AChE, and Pachymic acid and Dehydrotrametenolic acid were competitive inhibitors of AChE. Therefore, when different active ingredients are used in anti−Alzheimer’s disease drugs, the influence of the underlying environment on the inhibitory effect should be considered to achieve the best effect.

## 4. Materials and Methods

### 4.1. Apparatus

The analysis of HPLC products was carried out on a Waters 2695 coupled with a Waters 2998 Diode array detector, analysis SunFireTMC18 Column (250 mm × 4.6 mm, id 5 μm, Waters Corporation, Milford, MA, USA) and Sigma 1–14 centrifuge (Sigma, St. Louis, MO, USA) with ultramembrane filters (Microcon YM-30; Millipore, Billerica, MA, USA). A semi−preparative HPLC was performed using a 2545 Quaternary Gradient Module pump, a Fraction Collector III, a Waters 2489 UV/Vis detector, and a semi−prep SunFire^TM^C_18_ Column (100 × 19 mm, ID 5 μm, Waters Corporation, Milford, MA, USA). HSCCC was performed on a DE Spectrum HSCCC (Dynamic Extractions, Slough, UK). The kinetics of enzymatic reactions were managed using a High Performance Capillary Electrophoresis apparatus (Beckman Coulter, Krefeld, Germany).

### 4.2. Reagents and Materials

*Poria cocos* was purchased from Sichuan Province (China) and was identified by Professor Chunming Liu (Changchun Normal University, Changchun, China). AChE was obtained from Sigma (St. Louis, MO, USA). We also used a Microcon YM−30 (Millipore, Billerica, MA, USA) ultrafiltration chamber with the molecular weight cutoff of 30,000 Da. The phosphate−buffered saline (PBS) buffer was purchased from Sigma (St Louis, MO, USA). HPLC−grade acetonitrile and formic acid were purchased from Thermo Fisher Scientific (Waltham, MA, USA). Acetylthiocholine Chloride was purchased from Shanghai Yuanye Biotechnology Co., LTD. (CAS: 6050-81-3, Shanghai, China). Huperzine A was purchased from Shanghai Yuanye Biotechnology Co., LTD. (CAS: 102518−79−6, Shanghai, China). Chollne semiconductor solution was purchased from Shanghai Aladdin Biochemical Technology Co., LTD. (CAS: 123−44−1, Shanghai, China). The solvents and all other chemicals used in the study were of analytical grade and were purchased from Beijing Chemical Engineering Company (Beijing, China). Water was purified using a Milli−Q (Millipore, Boston, MA, USA) water purification system. 

### 4.3. Screening for Potential AChEIs

UF was used to screen the potential AChE inhibitors from the crude *P. cocos* extract. A PBS solution was used as the buffer solution for the AChE inhibitors. The reaction mixtures (200 μL) contained 90 μL of 0.5, 1.0, and 2.0 U/mL enzyme, 10 μL (20 mg/mL) sample, and 100 μL buffer solution, with a molecular mass of 30 kDa in 10 mM PBS solution buffer at 37 °C for 30 min. The control sample contained the same reaction mixture, but the sample was replaced with an equal amount of buffer solution. After incubation, each mixture was filtered through a YM−30 UF membrane with the molecular weight cutoff of 30 kDa using a centrifugal filter for 10 min, and the released active ingredients were identified using HPLC. The binding strength was used to characterize the binding strength of the compound and AChE. The calculation formula of the binding strength is illustrated in Equation (1). The *A_a_* and *A_b_* are the peak area of the blank and the experimental group, respectively.
(1)$$\mathrm{Binding}\;\mathrm{degree}=\frac{\left(A_{a}-A_{b}\right)}{A_{a}}\times 100\%$$

### 4.4. HPLC Conditions

We used a C_18_ (150 mm × 4.6 mm ID) analytical column, where acetonitrile and 0.1% phosphoric acid aqueous solution were used as mobile phases D and B, respectively, for gradient elution. The flow phase gradient program was as follows: 0~10 min, 55% D; 10~40 min, 55~100% D; 40~85 min, 100% D. The flow velocity, detection wavelength, sample volume, and column temperature were 0.4 mL/min, 242 nm, 15 μL, and 30 °C, respectively.

### 4.5. UPLC-Q-Exactive Conditions

UPLC conditions: Thermo analytical C_18_ column (100 × 2.1 mm ID); The mobile phase consisted of 0.1% formic acid water (A) −acetonitrile (D); Gradient elution: 0–5 min, 55% D; 5–15 min, 55~100% D; 15–30 min, 100% D; Volume flow rate: 0.5 mL/min; Column temperature: 35 °C; Detection wavelength: 242 nm; Injection volume: 3 μL.

Electrospray ionization (ESI)-MS experiments in negative mode were used for data collection. The following settings were applied to the instrument: capillary temperature: 350 °C; auxiliary gas flow rate: 5 bar; The full−scan of ions revealed molecules ranging in weight from 100 to 1500. The electrospray voltage of the ion source: 4.5 kV; sheath gas flow rate: 30 bar; capillary voltage: 35 V. A source collision−induced dissociation experiment was performed to acquire the detailed structural information. 

### 4.6. Molecular Docking Simulation Verification of Active Compounds and AChE

Autodock Vina 1.2.0 (The Scripps Research Institute, La Jolla, CA, USA) were used to simulate the docking mode between AChE and the active ingredients. The binding energies for all receptor and ligand conformations were calculated using the Lamarck Genetic Algorithm approach (LGA). Python 3.4 (DeLano Scientific LLC, Waren DeLano, CA, USA) was used to analyze the docking results of the active compounds and target proteins.

### 4.7. Separation of Active Ingredients from P. cocos

#### 4.7.1. HSCCC Separation Conditions

The following 16 sets of solvent systems were used to separate the active compounds in *P. cocos*. In order to investigate the *K* value of the iconic compound, we conducted pre-experiments on each solvent system. After the solvent systems were prepared, they were allowed to reach equilibrium. Subsequently, 2.0 mL of both the upper and lower phase was added to a test tube and 2.0 mg of *P. cocos* extract was added to the test tube and allowed to fully dissolve. The upper and lower phases of the obtained mixtures were filtered through a 0.45 μm filtration membrane and allowed to rest prior to HPLC detection.

After complete stratification, 1 mL of each upper and lower phase was taken and dried, and then 1 mL of methanol was dissolved for HPLC determination. The formula for calculating the distribution coefficient is shown below:
(2)$$K=\frac{A_{upper}}{A_{lower}}$$

In the above formula, *A*_upper_ is the peak area of the target compound in the upper phase, and the *A*_lower_ is the peak area of the target compound in the lower phase.

A nonlinear correlation was used to analyze the *K* values of the target compounds, and the optimal volume ratios were calculated. The *K* values of all compounds were in the range of 0.5−2.0, and thus the solvent systems were considered ideal, as illustrated in Table 2. Based on the obtained results, we selected PET : EtOAc : MeOH : H_2_O (4.0 : 1.0 : 3.0 : 2.0, *v/v/v/v*) as the solvent system. The other conditions of the HSCCC experiment were as follows: a detection wavelength of 254 nm, a mobile phase velocity of 1.5 mL/min, a screw speed of 800 rpm, a temperature of 25 °C, a sample volume of 5 mL, and a sample concentration of 50 mg/mL. 

#### 4.7.2. Semi−Preparative HPLC Separation Conditions

*P. cocos* extract (50 mg) was accurately weighed and dissolved in 5 mL of 100% MeOH solution followed by ultrasound-assisted filtration through a 0.45 μm filtration membrane.

The active components in the *P. cocos* extract were separated via semi-preparative HPLC, and the HPLC conditions (including the composition of mobile phase, flow, and injection volume) were optimized. The mobile phase typically consists of water, acetonitrile, and MeOH. Considering the polarity, viscosity, and separation effect, the acetonitrile and H_2_O solution were selected as mobile phases A and B, respectively. The detection wavelength and injection volume were 242 nm and 5 mL, respectively, and the gradient elution procedure is as follows: 10% A, 0~5 min; 10~100% A, 5–60 min; 100% A, 60~150 min.

### 4.8. Study on Enzymatic Reaction Kinetics of AChE Inhibitors

#### 4.8.1. Solution Preparation

Borax buffer solution: We precisely weighed 0.0381 g borax, dissolved it in deionized water, and kept it in a 100 mL volumetric flask to obtain a buffer solution with a concentration of 1.0 mmol/L (pH 6.80).

NaOH solution: We precisely weighed 0.04 g of the drug, dissolved it in 100 mL ultrapure water, and kept it in a 100 mL volumetric flask to obtain 0.1 mol/L NaOH solution for later use.

AChE solution: We accurately weighed 50 mg 200 U/g AChE and dissolved it in 5 mL 1 mmol/L borax buffer solution; thus, 2.0 U/L AChE solution was obtained. We stored the solution in the dark at −20 °C and diluted it with 1 mmol/L borax buffer solution before use.

Substrate ACh solution: We accurately weighed and dissolved 19.77 mg ACh in 5 mL 1.0 mmol/L borax buffer solution, and a substrate solution with a concentration of 20 mmol/L was obtained. We stored it in the dark at −20 °C and diluted it with 1 mmol/L borax buffer solution before use.

#### 4.8.2. Electrophoresis Method

The selected models were measured by the variation in the substrate ACh and product choline (Ch) in the enzyme reaction. The variations in ACh and Ch were measured using the capillary electroswimming method. The electroswimming strip was as follows: a capillary column temperature of 25 °C and a PDA detection wavelength of 230 nm. The washing sequence and timing of the capillary columns are listed in Table 3. 

## 5. Conclusions

In this study, a rapid screening method was established for the active compounds of anti-Alzheimer’s disease in *P. cocos*. We used core basic theories such as mathematical model, mass spectrometry, chromatography and computer molecular docking to comprehensively screen the active ingredients against AD in *P. cocos*. In doing so, we established the identification method of a compound structure based on LC−MS analysis and the separation method of active compounds with high efficiency, and rapid and successful recovery. We also explored the action mechanism of active compounds against AChE. This provides a theoretical basis for the targeted screening and isolation of active compounds and clinical drug development. We believe that this method can be used to identify bioactive compounds for the development of novel anti-Alzheimer’s disease therapeutics.

## Figures and Tables

**Figure 1 molecules-28-01228-f001:**
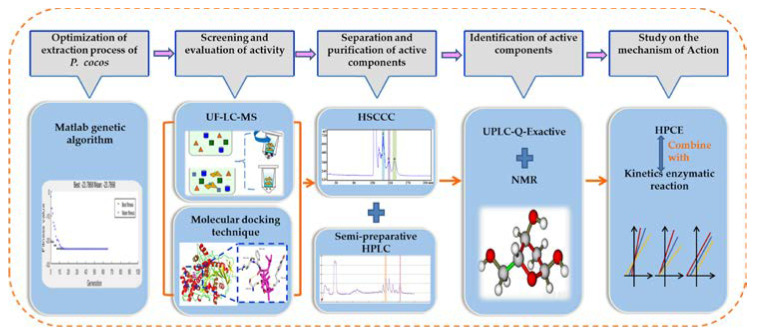
Schematic illustration of the workflow for exploring the activity screening, isolation of acetylcholinesterase inhibitors (AChEIs), and the in vitro inhibiting effect of *Poria cocos* (*P. cocos*).

**Figure 2 molecules-28-01228-f002:**
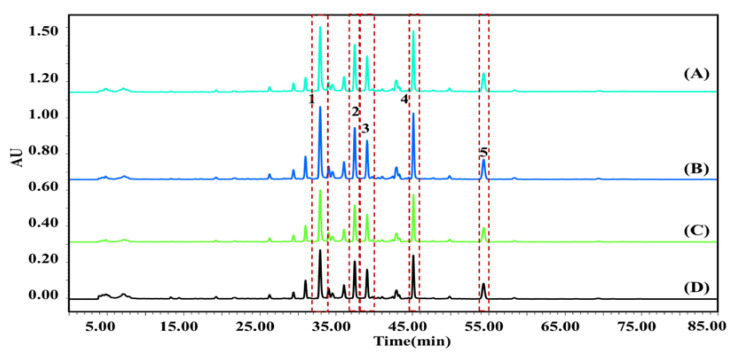
HPLC profiles of pure *P. cocos* extract (**D**) and extract combined with 0.5 (**A**), 1.0 (**B**), and 2.0 (**C**) U/mL acetylcholinesterase (AChE). (1) Tumulosic acid; (2) Polyporenic acid C; (3) 3-Epidehydrotumulosic acid; (4) Pachymic acid; (5) Dehydrotrametenolic acid.

**Figure 3 molecules-28-01228-f003:**
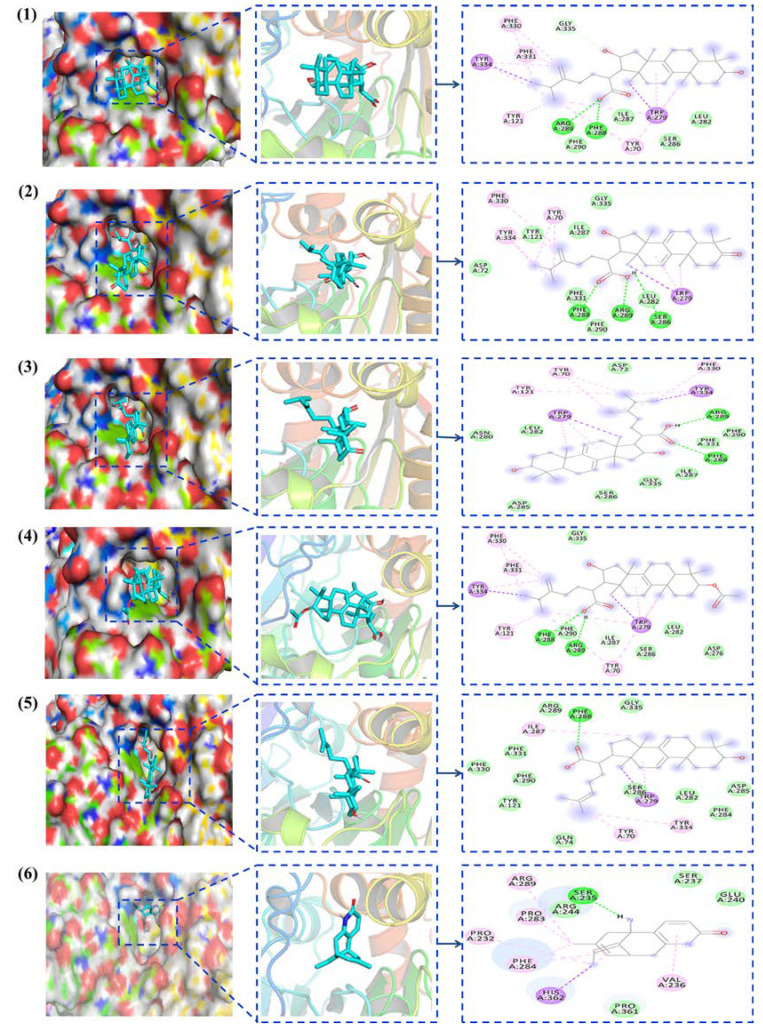
Molecular docking simulation of monomer active compounds and AChE. Compounds: (1) Tumulosic acid; (2) Polyporenic acid C; (3) 3−Epidehydrotumulosic acid; (4) Pachymic acid; (5) Dehydrotrametenolic acid; (6) Huperzine A.

**Figure 4 molecules-28-01228-f004:**
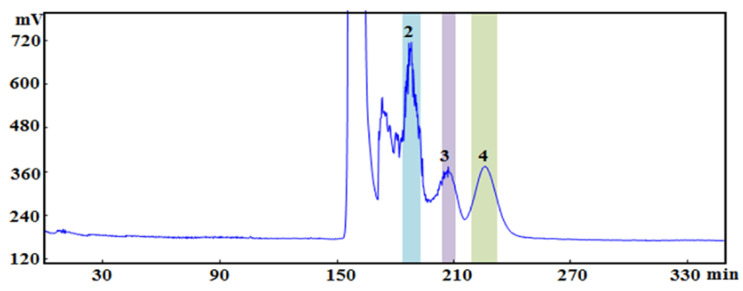
High−speed counter current chromatography (HSCCC) of the *P.cocos* extract. (solvent system: PET: EtOAc : MeOH : H_2_O (4.0:1.0:3.0:2.0, *v/v/v/v*); coiled column: 800 rpm; detection wavelength: 270 nm. Compounds: (2) Polyporenic acid C; (3) 3−Epidehydrotumulosic acid; (4) Pachymic acid).

**Figure 5 molecules-28-01228-f005:**
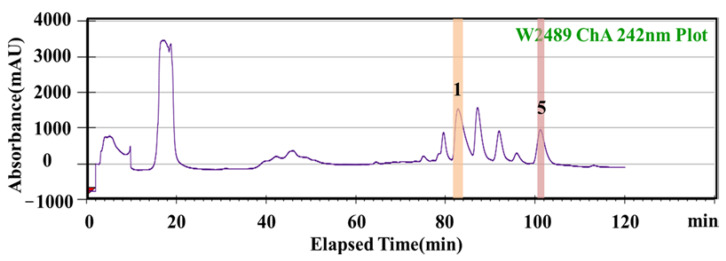
Semi−preparative high performance liquid chromatography (Semi−preparative HPLC) of *P.cocos* extract. (mobile phase acetonitrile (solvent A) and H_2_O (solvent C). Solvent gradient: 10% A, 0~5 min; 10~100% A, 5~60 min; 100% A, 60~150 min.; flow rate: 1.0 mL/min. Injection volume: 5.0 mL; detection wavelength: 270 nm. Compounds: (1) Tumulosic acid; (5) Dehydrotrametenolic acid.

**Figure 6 molecules-28-01228-f006:**
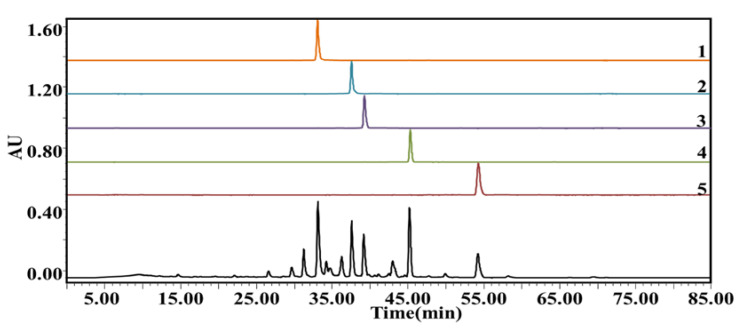
Chromatograms of AChEIs isolated from *P. cocos* crude extracts using HSCCC and semi−preparative HPLC. Compounds identified: (1) Tumulosic acid (98.11%); (2) Polyporenic acid C (96.29%); (3) 3−Epidehydrotumulosic acid (95.40%); (4) Pachymic acid (97.31%); (5) Dehydrotrametenolic acid (97.82%).

**Figure 7 molecules-28-01228-f007:**
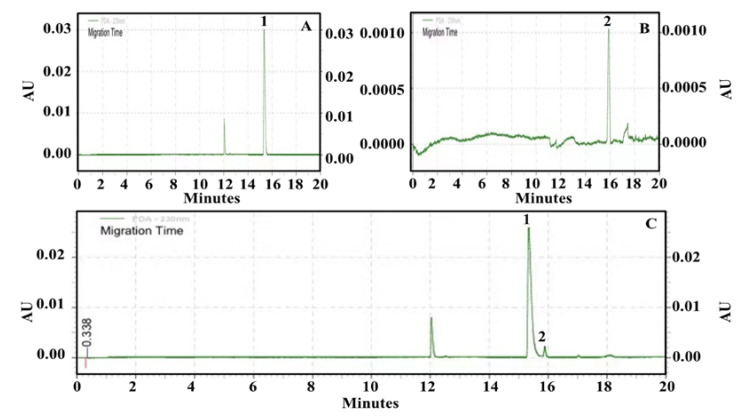
Capillary electrophoresis of acetylcholine (ACh) and choline (Ch) separation. (**A**) ACh standard (1) capillary electrophoresis diagram; (**B**) Capillary electrophoresis of Ch (2) standard; (**C**) Capillary electrophoresis after addition with AChE.

**Figure 8 molecules-28-01228-f008:**
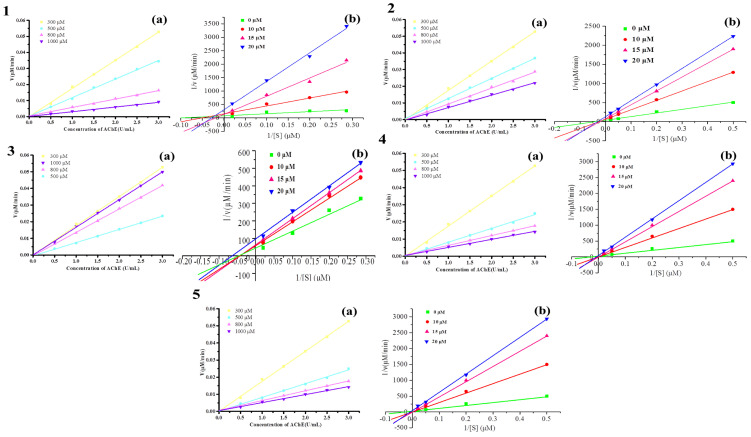
(1(**a**), 2(**a**), 3(**a**), 4(**a**), 5(**a**)) The measurement curve of inhibition type; (1(**b**), 2(**b**), 3(**b**), 4(**b**), 5(**b**)) Lineweaver−burk curve of the inhibitory effect on AChE of active compounds. (1) Tumulosic acid; (2) Polyporenic acid C; (3) 3−Epidehydrotumulosic acid; (4) Pachymic acid; (5) Dehydrotrametenolic acid.

**Table 1 molecules-28-01228-t001:** Retention time (t_R_) and negative electrospray ionization-mass spectra (ESI-MS) of the peak in UPLC.

Peak No.	t_R_ (min)	MS (Primary Mass Spectrometry) (neg.) *m/z*	MS^2^ (Secondary Mass Spectrometry) (neg.) *m/z*	Compound	Formula
1	12.137	485.2734	469.3123,467.2641, 439.1472, 423.2659, 337.0428	Tumulosic acid	C_31_H_50_O_4_
2	15.683	481.3371	463.4434, 437.2927, 421.3485, 403.3042	Polyporenic acid C	C_31_H_45_O_4_
3	17.562	483.7835	467.0731, 465.2334, 439.1454, 437.2517, 421.2265, 405.1832, 337.1517	3-Epidehydrotumulosic acid	C_31_H_48_O_4_
4	22.461	527.3791	509.1385, 481.2173, 482.8624, 467.2475, 465.1022, 449.1631	Pachymic acid	C_33_H_52_O_5_
5	26.330	453.3451	435.2734, 393.1805, 371.0163	Dehydrotrametenolic acid	C_30_H_46_O_3_

**Table 2 molecules-28-01228-t002:** Partition coefficient (*K*) values of isoflavones from *P. cocos* in different two-phase solvent systems.

No.	Solvent System	*v/v/v/v*	*K* _1_	*K* _2_	*K* _3_	*K* _4_	*K* _5_
1	HEX:EtOAc:MeOH:H_2_O	3.0:6.0:4.0:2.0	0.53	0.92	0.68	1.43	0.81
2	HEX:EtOAc:MeOH:H_2_O	5.0:1.0:5.0:1.0	——	0.01	——	0.04	——
3	HEX:EtOAc:MeOH:H_2_O	6.0:2.0:5.0:1.0	0.02	0.06	0.03	1.35	0.02
4	HEX:EtOAc:MeOH:H_2_O	7.0:3.0:2.0:8.0	0.30	0.36	0.33	0.39	0.20
5	PET:EtOAc:MeOH:H_2_O	1.5:0.5:1.3:0.8	0.25	0.62	0.35	0.94	1.76
6	PET:EtOAc:MeOH:H_2_O	1.0:0.5:1.3:0.8	0.16	0.56	0.24	1.30	0.95
7	PET:EtOAc:MeOH:H_2_O	4.0:1.0:3.0:2.0	0.63	0.70	0.58	0.92	1.31
8	PET:EtOAc:MeOH:H_2_O	5.0:1.0:5.0:2.0	0.01	0.03	0.01	0.09	0.04
9	HEX:MeOH:H_2_O	2.0:1.0:1.0	0.04	——	0.03	0.06	0.07
10	HEX:MeOH:H_2_O	3.0:2.0:4.0	0.02	0.03	0.02	0.02	0.13
11	HEX:Ethanol:H_2_O	2.0:1.0:5.0	0.91	0.88	0.99	1.03	0.94
12	HEX:Ethanol:H_2_O	3.0:3.0:5.0	——	0.03	0.01	0.02	0.03
13	EtOAc:MeOH:n-BuOH:H_2_O	5.0:2.0:5.0:15.0	11.56	12.25	12.56	10.79	——
14	EtOAc:MeOH:n-BuOH:H_2_O	5.0:2.0:2.5:12.0	——	——	——	——	——
15	EtOAc:MeOH:n-BuOH:H_2_O	6.0:2.0:1.0:10.0	15.38	22.49	18.06	18.65	——
16	EtOAc:n-BuOH:H_2_O	5.0:1.0:8.0	7.56	10.47	8.25	12.69	6.99

**Table 3 molecules-28-01228-t003:** Washing sequence and schedule of the capillary column.

	Time (min)	Event	Value	Duration	Inlet Vial	Outlet Vial	Summary
1		Rinse-Pressure	20.0 psi	3.00 min	BI^1^:A1^3^	BO^2^:A2^4^	forward
2		Inject-Pressure	0.5 psi	5.0 s	BI:E1^5^	BO:A2	Override, forward
3		Inject-Pressure	0.5 psi	5.0 s	BI:F2^6^	BO:A2	No override, forward
4	0.00	Separate-Voltage	10.0 kV	20.00 min	BI:A1	BO:A1	10.00 Min ramp, normal polarity
5	20.00	End					

^1^BI: buffer solution inject; ^2^BO: buffer solution outlet; ^3^A1, ^4^A2, ^5^E1, ^6^F2: Sample bottle location.

## Data Availability

The datasets used and analyzed during the study are available from the corresponding author upon reasonable request.

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
