# Peer review of "Efficient Combination of Complex Chromatography, Molecular Docking and Enzyme Kinetics for Exploration of Acetylcholinesterase Inhibitors from Poria cocos"

_molecules, 2023, doi:10.3390/molecules28031228_

Round 1

Reviewer 1 Report

The present paper entitled “Efficient combination of complex chromatography, molecular docking and enzyme kinetics for exploration of acetylcholinesterase inhibitors from Poria cocos” by Tong Wu and others describes different methods used to test the activity of Poria cocos compounds to treat Alzheimer’s disease by inhibiting acetylcholinesterase enzyme. Thus, my current decision is "major revision". However, before it is accepted for publication the concerns raised below need to be properly addressed.

1-      Although the manuscript is generally moderately written, but careful revision by a language expert would be helpful to improve readability.

2-      The quality of all figures used must be improved.

3-      The docking study was not well explained since they did not mention the position and the size of the grid box that has been used.

4-      I would suggest that you redock (redocking study) the most active compounds with a standard (you can use any available acetylcholinesterase inhibitors).

5-      The paper is overly full as some parts in the materials and methods section are supposed to be migrated to the supplementary materials to give more space for your results.

Reviewer 2 Report

This paper explores the isolation and testing of bioactive compounds from Chinese medicinal plants. It is a nice body and work but it was difficult to follow. A few comments-

A simple explanation of what the ultrafiltration step actually involves (L79 pg2) without having to check other literature. The results section (2.2.1) was very brief. As someone with little experience in this technique it was difficult to follow. Figure 2 (in black and white) is unreadable and barely better in colour- I can see that there is an increased absorbance (Area under curve) for the 5 compounds . Is there a better way to show this data? In equation 1 what do the subscripts a, b and c represent? The text labels the compounds 1-5 but the figure gives the name. Are the compounds identified in this step by MS? Again not clear but I assume so as the next step is docking to validate the UF step...

If these are known compounds where have they been identified before?

A figure showing their chemical structure would have been useful. A discussion of the key binding interactions (docking) maybe?

What was the PDB code used for the enzyme in the docking?

A comment on where these compounds were found to dock? They all look to be in a different location on the enzyme. (It is hard to tell as three of the figure show the enzyme in the same sort of pose, the other two are different). Where does the native ligand bind in comparison? Does this correlate to the kinetics where some were found to be competitive and non competitive?

Here the authors use both HSCCC and prep HPLC to isolate the compounds. It is not clear why both have been done. Both techniques gave compounds with purity >95%. Why were only compound 2,3,and 4 isolated by HSCCC and only compounds 1 and 5 by HPLC. Could compounds 2,3 and 4 also have been isolated using prep HPLC? There are at least three unlabelled peaks in figure 5? A thorough explanation of the advantages and disadvantages of each technique would have been useful.

A comment of activity based on 'combined strength capabilities' from the UFLCMS step, docking binding energies and finally the actual inhibition studies. The report states both the UF and docking data is the same, however what about the inhibition studies?

·        Compounds are sometimes numbered, sometimes given a letter code (eg Figure 3) be consistent

·        Starting a sentence with therefore (couple of times) or and (5. Conclusions..)

·        What is the error margin- going to a second decimal point for percentage calculations seems overboard.

·        Line 54 missing a space

·        Line 113 a space between times and The (with no capital)

·        Line 160 purities of four compounds but there are only three

·        Line 376- weigh 0.08 g of the drug?

Round 2

Reviewer 1 Report

The author seems to respond properly to my comment, but I still confused regarding the docking study. I strongly suggest the authors to do the redocking study again (validation of molecular docking) and try to do it properly because it will help the reader to understand whether these active compounds bind to the active site of the enzyme or another site. Also, could the authors explain this sentence and what was the method that has been used to support this conclusion? “This provideds a theoretical basis for the study of metabolism of active compounds in vivo and in clinical drug development.”
